# MicroRNA-27a-5p Downregulates Expression of Proinflammatory Cytokines in Lipopolysaccharide-Stimulated Human Dental Pulp Cells via the NF-κB Signaling Pathway

**DOI:** 10.3390/ijms25179694

**Published:** 2024-09-07

**Authors:** Shihan Wang, Nobuyuki Kawashima, Peifeng Han, Keisuke Sunada-Nara, Ziniu Yu, Kento Tazawa, Mayuko Fujii, Thoai Quoc Kieu, Takashi Okiji

**Affiliations:** 1Department of Pulp Biology and Endodontics, Graduate School of Medical and Dental Sciences, Tokyo Medical and Dental University (TMDU), Tokyo 113-8549, Japan; wang.endo@tmd.ac.jp (S.W.); hpfendo@tmd.ac.jp (P.H.); k.nara.endo@tmd.ac.jp (K.S.-N.); yu.endo@tmd.ac.jp (Z.Y.); kenendo@tmd.ac.jp (K.T.); neigedormir@gmail.com (M.F.); t.okiji.endo@tmd.ac.jp (T.O.); 2Department of Pediatric Dentistry, Faculty of Odonto-Stomatology, University of Medicine and Pharmacy at Ho Chi Minh City, Ho Chi Minh City 17000, Vietnam; kieuquocthoai@ump.edu.vn

**Keywords:** microRNA-27a-5p, human dental pulp cell, NF-κB signaling pathway, proinflammatory cytokine, pulpal inflammation

## Abstract

MicroRNA-27a-5p (miR-27a-5p) was significantly upregulated in dental pulp inflammation, yet its underlying mechanisms remain unclear. This study investigated the effect of miR-27a-5p on the expression of proinflammatory cytokines in human dental pulp cells (hDPCs) stimulated by lipopolysaccharide (LPS). LPS-stimulated hDPCs showed concurrent increases in the expression of miR-27a-5p and proinflammatory cytokines (IL-6, IL-8, and MCP1), and the increased expression was suppressed by NF-κB inhibitor BAY 11-0785. Transfection of the miR-27a-5p mimic downregulated the expression of proinflammatory cytokines, NF-κB activity, and the expression of NF-κB signaling activators (TAB1, IRAK4, RELA, and FSTL1) in LPS-stimulated hDPCs. Luciferase reporter assays revealed that miR-27a-5p bound directly to the 3’-UTR of TAB1. siTAB1 downregulated NF-κB activity and proinflammatory cytokine expression. Downregulation of proinflammatory cytokine expression, NF-κB activity, and NF-κB signaling activator expression (TAB1, IRAK4, and RELA) was also found in LPS-stimulated rat incisor pulp tissue explants following transfection with the miR-27a-5p mimic ex vivo. MiR-27a-5p, whose expression was induced by NF-κB signaling, negatively regulated the synthesis of proinflammatory cytokines via targeting NF-κB signaling. In particular, TAB1, a potent NF-κB activator, was targeted by miR-27a-5p. These results provide insights into the negative regulatory effects of miR-27a-5p, particularly those targeting the TAB1-NF-κB signaling pathway, on pulp inflammation.

## 1. Introduction

Pulpal inflammation is predominantly induced by dental caries-associated bacteria and their related products. Local immune responses in this tissue manage these microbiological stimuli [1]. Dental pulp cells close to infected dentinal tubules are the first defense against bacterial stimuli. Prolonged bacterial invasion into pulp tissue induces the infiltration of specific immunocompetent cells, such as macrophages, dendritic cells, and lymphocytes [2]. Odontoblasts and dental pulp fibroblasts express Toll-like receptor (TLR) 4 and TLR2 [3,4], which function as pattern-recognition receptors that bind to lipopolysaccharide (LPS), a typical pathogen-associated molecular pattern [5]. Stimulation of TLR2/4 by LPS activates several signaling pathways, including nuclear factor-κB (NF-κB), in dental pulp cells and subsequently induces the synthesis of proinflammatory mediators, including interleukin (IL)-1β, tumor necrosis factor α (TNFα), IL-6, IL-8, and monocyte chemotactic protein 1 (MCP1) [5,6,7,8]. These infection-induced inflammatory molecules are involved in eradicating the invading microbes, but, concurrently, they have the potential to induce collateral tissue damage [9]. Excessive or longstanding inflammatory reactions can induce severe pulp tissue damage that may result in total pulp necrosis. Conversely, a relatively mild inflammatory response encourages tissue regeneration by activating dental pulp stem cell (DPSC) migration, proliferation, and differentiation [10,11,12]. Low-grade inflammation might play a pivotal role in initiating early reparative or regenerative processes [13]. Therefore, it is important to regulate the inflammatory and immune responses of dental pulp at an appropriate level.

MicroRNAs (miRNAs) are a class of non-coding RNAs of approximately 23 nucleotides and play crucial roles in the negative regulation of post-transcriptional gene expression. They bind to the 3′-untranslated region (UTR) to induce degradation of target mRNAs and impede translation [14,15]. MiRNAs have an important effect on various biological processes, such as cell proliferation, survival, apoptosis, and inflammation [16,17] Increasing evidence has revealed that the expression levels of certain miRNAs vary during inflammatory responses [18]. In human leukemic cells, miR-146a is upregulated upon LPS binding to TLR 2/4 and subsequent NF-κB activation, and the promoted miR-146a downregulates proinflammatory cytokines by targeting activators of NF-κB signaling, such as tumor necrosis factor receptor-associated factor 6 (TRAF6) and IL-1 receptor-associated kinase 1 (IRAK1) [19]. A reduction in miR-223 is presumably involved in the promotion of irreversible pulpitis in human dental pulp tissue because miR-223 negatively regulates the production of IL-1β and IL-18 via the miR-223/NLRP3/CASP1 axis in human dental pulp fibroblasts [20]. Moreover, miR-27a has a critical role as an anti-inflammatory mediator in acute lung injury [21], the spinal cord [22], human articular chondrocytes [23], and human aortic endothelial cells [24] by targeting multiple genes. Our previous study evaluated LPS-induced miRNA expression changes in human dental pulp cells (hDPCs) compared to non-stimulated cells using a miRNA array and found that miR-27a-5p was among highly upregulated miRNAs with a 3.14-fold increase (Appendix A). However, much remains to be clarified concerning the regulatory role of miR-27a-5p in dental pulp inflammation. Therefore, this study investigated how miR-27a-5p regulates proinflammatory cytokine expression in LPS-stimulated hDPCs, focusing on the LPS/TLR4/NF-κB pathway as the intracellular signaling responsible for miR-27a-5p-induced regulation.

## 2. Results

### 2.1. LPS/NF-κB Signaling Promotes miR-27a-5p Expression and Proinflammatory Mediator Levels in LPS-Stimulated hDPCs

The application of LPS significantly increased the expression level of miR-27a-5p at 4 h in hDPCs (*p* < 0.001; Figure 1a). In experimentally-induced rat pulpitis tissue, hematoxylin and eosin (H&E) staining indicated a typical inflammatory reaction characterized by blood vessel dilation and inflammatory cells infiltration (Figure 1b). In vivo results corroborated a significant elevation in miR-27a-5p expression following 6 h of LPS exposure (*p* < 0.001, Figure 1c). Moreover, the mRNA expression levels of *IL-6*, *IL-8*, and *MCP1* in hDPCs were significantly increased in response to LPS stimulation (*p* < 0.01–0.0001; Figure 1d). LPS significantly promoted NF-κB activity in hDPCs (*p* < 0.001; Figure 1e), which was effectively blocked by BAY 11-7085 (*p* < 0.05; Figure 1e). The application of BAY 11-7085 significantly downregulated LPS-induced *IL-6, IL-8*, and *MCP1* expression (*p* < 0.05 or 0.01; Figure 1f) and LPS-induced miR-27a-5p expression in hDPCs (*p* < 0.01; Figure 1g).

### 2.2. MiR-27a-5p Downregulates Synthesis of Proinflammatory Mediators via the NF-κB Pathway in LPS-Stimulated hDPCs

In hDPCs stimulated by LPS, transfection of the miR-27a-5p mimic induced a significant reduction in NF-κB p65 (RELA) phosphorylation (*p* < 0.05; Figure 2b), NF-κB activity (*p* < 0.01; Figure 2c), and the mRNA/protein expression levels of IL-6, IL-8, and MCP1 (*p* < 0.05–0.0001; Figure 2d,e).

### 2.3. MiR-27a-5p Targets TAB1, IRAK4, and RELA in NF-κB Signaling

Candidate miR-27a-5p targets were searched for among NF-κB signaling molecules using TargetScanHuman https://www.targetscan.org/vert_80/ (accessed on 5 September 2024), and TGF-beta activated kinase 1 binding protein 1 (TAB1), IRAK4, NF-κB p65 (RELA), and follistatin-like 1 (FSTL1) [25] were found. The forced expression of miR-27a-5p markedly downregulated the mRNA/protein levels of TAB1 (*p* < 0.001), IRAK4 (*p* < 0.05), RELA (*p* < 0.01), and FSTL1 (*p* < 0.05) in LPS-stimulated hDPCs (Figure 3a,b,d,e,g,h,j). Immunofluorescence staining confirmed the suppressive effects of miR-27a-5p on the expression of TAB1 (Figure 3c), IRAK4 (Figure 3f), and RELA (Figure 3i) in LPS-stimulated hDPCs. MiR-27a-5p has been reported to interact with the 3ʹ-UTR of IRAK4, RELA, and FSTL1, but not TAB1 [23,26,27], and therefore, we next conducted luciferase reporter assays of TAB1 in miR-27a-5p-overexpressing hDPCs. Transfection of hDPCs with reporter vectors, each of which contained the miR-27a-5p target sequences in the TAB1 3’-UTR, resulted in downregulation of their luciferase activities (*p* < 0.001 and 0.05; Figure 3k,l). These effects were canceled by the transfection of a reporter vector encoding mutant miR-27a-5p target sites in the TAB1 3ʹ-UTR (*p* > 0.05; Figure 3k,l).

Next, the evaluation of TAB1 depletion using small interfering RNA against TAB1 (si-TAB1) demonstrated the effective knockdown of TAB1 expression (*p* < 0.01; Figure 4a,b). si-TAB1 significantly downregulated phosphorylation of NF-κB p65 (RELA) (*p* < 0.05; Figure 4c) and the expression levels of *IL-6* and *IL-8* (*p* < 0.05 or 0.01; Figure 4d).

### 2.4. MiR-27a-5p Exerts a Downregulatory Effect on NF-κB Activity and Proinflammatory Cytokine Expression in LPS-Stimulated Rat Pulp Tissue Ex Vivo

The overexpression of miR-27a-5p (Figure 5a) led to downregulation of NF-κB p65 (RELA) phosphorylation (*p* < 0.05; Figure 5b) and *Il6* and *Mcp1* mRNA levels (*p* < 0.05 or 0.01; Figure 5c). Moreover, the overexpression of miR-27a-5p induced significant decreases in the expression of Tab1, Irak4, and Rela at mRNA and protein levels (*p* < 0.05–0.001; Figure 5d–l).

In summary, our findings show that in LPS-stimulated hDPCs, the NF-κB signaling pathway induces miR-27a-5p expression. This, in turn, is negatively regulated by miR-27a-5p itself through the downregulation of TAB1, IRAK4, RELA, and FSTL1 (Figure 6).

## 3. Discussion

Elevated miR-27a expression after exposure to LPS has been reported in rat pancreatic acinar cells [28], dairy cow mammary epithelial cells [29], and CD-1 mice [30]. However, few studies have investigated the role of miR-27a in pulpal inflammation. In this study, we revealed significant upregulation of miR-27a-5p in hDPCs and rat incisor pulp tissues stimulated by LPS. LPS stimulation also upregulated the expression of proinflammatory cytokines, including *IL-6*, *IL-8*, and *MCP1*, in hDPCs, which occurred concurrently with the increase in miR-27a-5p expression. These findings indicate a shared upstream signaling pathway between miR-27a-5p and proinflammatory cytokines. TLRs, which are regarded as pattern recognition receptors, are crucial for immune responses, especially pathogen recognition [31]. In particular, TLR2 and TLR4, expressed on human dental pulp cells [32,33,34,35,36], are essential for LPS detection, which is involved in triggering the transcription factor NF-κB, leading to the synthesis of various proinflammatory mediators [31,36]. We verified that activation of NF-κB and its downstream mediators induced by LPS was negatively regulated in hDPCs by treatment with BAY 11-7085, an NF-κB inhibitor. Furthermore, LPS-stimulated miR-27a-5p expression was downregulated by BAY 11-7085, indicating that NF-κB signaling is upstream of both miR-27a-5p and proinflammatory mediators, as reported previously [37,38]. The binding site of NF-κB p65 (RELA) is located in the miR-27a promoter region [37,38].

We investigated whether miR-27a-5p produced in LPS-stimulated hDPCs and pulp tissues exerts on the synthesis of proinflammatory mediators. The results revealed that overexpression of miR-27a-5p downregulated the expression of IL-6, IL-8, and MCP1 at both mRNA and protein levels in LPS-stimulated hDPCs and the expression of *Il6* and *Mcp1* mRNAs in LPS-applied rat pulp explants. These findings demonstrate miR-27a-5p as a negative regulator of proinflammatory cytokine synthesis, and it may act as a member of negative feedback loops. We have reported that miR-21 and miR-146b negatively regulate proinflammatory cytokine synthesis by targeting NF-κB in LPS-stimulated hDPCs [39,40], and miR-27a-5p may be a negative regulator in pulpal inflammation. Excessive synthesis of proinflammatory cytokines leads to the destruction of homeostasis, which induces severe damage to host tissues [41,42,43]. Taken together, it is reasonable to propose that miR-27a-5p contributes to negative feedback networks in inflamed dental pulp to maintain the integrity of this tissue.

MiRNA regulates target genes through partial or complete complementary binding between the target gene 3ʹ-UTR and the miRNA seed base, suppressing target gene expression and subsequent protein synthesis [44]. MiRNA-27a modulates multiple components of the TLR/NF-κB pathway, such as IRAK4, RELA [24,26], and FSTL1 [27]. In this study, expression of these genes was attenuated through overexpression of miR-27a-5p in LPS-stimulated hDPCs and rat incisor pulp tissues. We predicted TAB1 to be a candidate target of miR-27a-5p among NF-κB pathway components. TAB1 is required to initiate TGF-beta-activated kinase 1 (TAK1) activation, triggering several downstream inflammatory signaling pathways, including NF-κB, and inducing the synthesis of proinflammatory cytokines. Activation of TAK1 does not occur in TAB1^−/−^ mouse embryonic fibroblasts. The TAK1-TAB complex is reported to be closely related to multiple diseases [45,46]. Knocking down TAB1 reduces cell hypertrophy of cardiomyocytes [47]. Deficiency or mutations of TAB1 attenuate inflammation reactions induced by LPS, TNFα, or IL-1β stimulation [46,48,49]. Expression of TAB1 was downregulated by miR-27a-5p in LPS-stimulated hDPCs and rat incisor pulp tissues. The luciferase reporter assays further revealed that the 3ʹ-UTR region of TAB1 contained the miR-27a-5p binding sequence. Knockdown of TAB1 led to decreased levels of phosphorylated NF-κB and IL-6, IL-8, and MCP1. Collectively, it was demonstrated that depletion of TAB1 significantly diminished NF-κB activation and the inflammatory response induced by LPS stimulation. These results provide a new insight into the regulatory mechanisms of miR-27a-5p in inflammatory reactions, particularly those involving TAB1, where miR-27a-5p downregulates the production of proinflammatory mediators by blocking the TAB1-NF-κB signaling pathway.

MiRNAs are being explored as therapeutic agents in cancer and various other fields [50]. Exosomes, which are small vesicles released from cells, are under investigation as carriers for miRNA delivery due to their ability to modulate immune responses and target specific disease mechanisms [51]. MiRNAs like exosomal miR-21 in hDPCs, colon cancer, and miR-15 and miR-16 in acute myeloid leukemia are promising therapeutic targets [50,51]. Research also extends to lung cancer, retinal disorders, and toxicogenomics [50]. Despite challenges such as rapid degradation and limited tissue penetration, clinical/preclinical trials are showing positive results [50]. Our observations highlight the prospective application of miR-27a-5p as a treatment approach for pulp inflammation.

However, a limitation of this study lies in the absence of validity for miR-27a-5p detection in human pulpal inflammation in vivo. Further research is required to validate the effectiveness and safety of miR-27a-5p as a viable in vivo therapy, despite the positive results of ex vivo evaluations using rat dental pulp tissue and in vitro investigations using hDPCs. Moreover, additional studies are needed to determine the optimal methodologies to apply miR-27a-5p to inflamed human pulp tissue.

## 4. Materials and Methods

### 4.1. Cell Culture and Treatments

The use of hDPCs was approved by the Ethics Committee of the Tokyo Medical and Dental University (D2014-039). hDPCs harvested from pulp tissue of wisdom teeth were cultured in α minimum essential medium (FUJIFILM Wako Pure Chemical, Osaka, Japan) containing 10% fetal bovine serum (Thermo Fisher Scientific, Waltham, MA, USA) and 1% penicillin/streptomycin (FUJIFILM Wako Pure Chemical) at 37 °C with 5% CO_2_. Hsa-miR-27a-5p mimic (miR-27a-5p mimic; Thermo Fisher Scientific) was transfected into cells using Lipofectamine RNAiMax (Thermo Fisher Scientific). As a control, mirVana miRNA mimic Negative Control #1 (NC; Thermo Fisher Scientific) was employed. si-TAB1 (Thermo Fisher Scientific) was transfected into hDPCs using Lipofectamine 3000 (Thermo Fisher Scientific). hDPCs were stimulated with 100 ng/mL LPS (*Escherichia coli* O111:B4; Sigma-Aldrich, St. Louis, MO, USA). To inhibit NF-κB signaling, 1 µM BAY 11-7085 (Cayman Chemical, Ann Arbor, MI, USA) was employed.

### 4.2. Establishment of the Pulpitis Model in Rats

Animal experiments were approved by the Animal Experimentation Committee of the Tokyo Medical and Dental University (A2019-297C). Six-week-old male Sprague-Dawley rats (Clea Japan, Tokyo, Japan) were fed a standard laboratory rat diet and allowed to drink water freely. The rats were anesthetized with 50 mg/kg ketamine hydrochloride (Parke-Davis, Detroit, MI, USA) and 20 mg/kg xylazine (Bayer, Leverkusen, Germany) injected intraperitoneally. Then, pulp tissue in maxillary and mandibular incisors was exposed using a round diamond bur, followed by application of 1 μL LPS in sterile saline (10 mg/mL) using sterile paper points. Sterile saline served as a negative control. The cavities were sealed with a light-curing resin (G-FIX; GC, Tokyo, Japan). The rats were euthanized by CO_2_ inhalation at 0 (no pulp exposure), 3, 6, 12, and 24 h postoperatively. Following tooth extraction, extirpated pulp tissues were stored in RNAlater (Thermo Fisher Scientific) and subsequently analyzed for miR-27a-5p expression by reverse transcription-quantitative polymerase chain reaction (RT-qPCR). Rats were euthanized with CO_2_ at 0 and 12 h for histological analysis. Upper and lower incisors were harvested, fixed in 4% paraformaldehyde in PBS at 4 °C for 24 h, and demineralized with EDTA for 14 days. The samples were then embedded in a tissue-embedding medium (Tissue-Tek O.C.T. Compound; Sakura Finetek, Tokyo, Japan) and sectioned into 10 μm slices using a cryostat (CM3050; Leica Microsystems, Wetzlar, Germany). These samples were stained with H&E.

### 4.3. Ex Vivo miR-27a Mimic Transfection

Male Sprague-Dawley rats (6 weeks old; Clea Japan) were euthanized by CO_2_ inhalation, and their maxillary and mandibular incisor pulp tissues were removed. The pulp tissues were transfected with mirVana miRNA mimic miR-27a-5p or mirVana miRNA mimic Negative Control #1 (Thermo Fisher Scientific) using Lipofectamine RNAiMAX. Twenty-four hours after transfection, the pulp tissues were stimulated ex vivo with 200 ng/mL LPS for 4 h. Then, the pulp tissues were rinsed with phosphate-buffered saline (PBS) and fixed in 4% paraformaldehyde at 4 °C overnight. They were embedded in O.C.T. Compound and cryosectioned at 10 μm for immunofluorescence analysis.

### 4.4. Reverse Transcription-Quantitative Polymerase Chain Reaction

The QuickGene-Mini80 nucleic acid isolation system (FUJIFILM Wako Pure Chemical) was used to extract total RNA. Reverse transcriptase (RevertAid H Minus Reverse Transcriptase, Thermo Fisher Scientific) and PrimeScript™ RT Master Mix (Takara Bio, Kusatsu, Japan) were used to synthesize cDNA and perform RT-qPCR. mRNA detection was performed using a real-time PCR detection system (CFX 96, Bio-Rad, Hercules, CA, USA). For miR-27a-5p, total RNA was extracted with a mirVana miRNA Isolation Kit (Thermo Fisher Scientific), and cDNA was synthesized using RT primers specific for miR-27a-5p and U6 in TaqMan microRNA Assays (Thermo Fisher Scientific) using a microRNA Reverse Transcription Kit (Thermo Fisher Scientific). For real-time PCR, the CFX 96 was used with a TaqMan Universal Master Mix II, no UNG (Thermo Fisher Scientific). The formula 2^−ΔΔCt^ was used to calculate relative gene expression with ACTB or U6 as internal controls. Table 1 shows the primer sequences.

### 4.5. Western Blotting

Protein samples were extracted using radioimmunoprecipitation buffer containing inhibitors of proteases (cOmplete, Sigma-Aldrich) and phosphatases (PhosSTOP, Sigma-Aldrich). After mixing with loading buffer and denaturation at 95 °C for 3 min, samples were subjected to electrophoresis in polyacrylamide gels containing sodium dodecyl sulfate (e-PAGEL; ATTO, Tokyo, Japan). Separated proteins were transferred onto polyvinylidene difluoride membranes (Immobilon-P; Merck Millipore, Burlington, MA, USA). The membranes were reacted with the following primary antibodies: rabbit anti-TAB1 (1:300, Q15750, polyclonal, rabbit; Proteintech, Rosemont, IL, USA), rabbit anti-IRAK4 (1:300, Q69FE3, polyclonal, rabbit; Proteintech), rabbit anti-NF-κB p65 (RELA, 1:1000, D14E12, monoclonal, rabbit; Cell Signaling Technology, Danvers, MA, USA), and anti-glyceraldehyde-3-phosphate dehydrogenase (GAPDH, 1:4000, PM053-7; Medical and Biological Laboratories, Nagoya, Japan). As a secondary antibody, horseradish peroxidase-conjugated anti-rabbit IgG (1:5000, W4011; Promega, Tokyo, Japan) was used. Following incubation with a chemiluminescent horseradish peroxidase substrate (Immobilon, Millipore, Burlington, MA, USA), images of bands were acquired with the LAS-3000 mini-imaging system (Fujifilm, Tokyo, Japan). A densitometry-based approach was used to measure pixel density with ImageJ software (version 2; National Institutes of Health, Bethesda, MD, USA), and band density ratios were calculated.

### 4.6. Cytometric Bead Array

The culture supernatant of hDPCs was collected for analysis. The Human Inflammation Standard (BD Biosciences, Franklin Lakes, NJ, USA) was reconstituted using Assay Diluent (BD Biosciences) and then mixed with Human Inflammation Capture Bead suspension for IL-6, IL-8, and MCP1 (BD Biosciences) diluted in Bead Capture Diluent (BD Biosciences). Cytokine detection was performed using phycoerythrin-conjugated antibodies. Quantification was based on a standard curve plot. Fluorescence from the phycoerythrin-conjugated antibodies was detected with a FACS Canto II flow cytometer (BD Biosciences), following the manufacturer’s protocol. Data analysis was performed using FCAP Array Software v3.0 (BD Biosciences).

### 4.7. Luciferase Assays Using NF-κB and TAB1 3′-UTR Reporter Vectors

The pGL4.32 vector (luc2P/NF-κB-RE/Hygro, Promega), incorporating five copies of the NF-κB response element, was employed to assess luciferase activity associated with the NF-κB signaling pathway. The reporter vector and either the miR-27a-5p mimic or miR-27a-5p negative control (NC) were transfected into hDPCs using Lipofectamine 3000. hDPCs were subsequently exposed to 100 ng/mL LPS.

For the luciferase assay with the TAB1 3′-UTR reporter vector, insertion of synthesized TAB1 3′-UTRs containing wildtype or mutated hsa-miR-27a-5p target sequences (400 bp each; Eurofins Genomics, Ebersberg, Germany) was carried out in XhoI and HindIII sites of the pMIR-REPORT vector (Thermo Fisher Scientific). Using Lipofectamine 3000, these vectors were co-transfected into hDPCs with the miR-27a-5p mimic or NC. The cells were lysed using luciferase cell culture lysis reagent. A luminometer (Luminescence PSN, Atto) and luciferase assay equipment (Promega) were used to measure luciferase activity.

### 4.8. Immunofluorescence

hDPCs were rinsed with PBS and fixed in 4% paraformaldehyde at 4 °C overnight. The fixed cells and rat pulp tissue sections were reacted with primary rabbit antibodies against ΤΑΒ1 (1:500), ΙRAΚ4 (1:500), and RELA (1:500) at 4 °C overnight. The specimens were then incubated with Alexa Fluor 488-conjugated anti-rabbit IgG (1:500, donkey; Abcam, Cambridge, UK) and mounted using Fluoroshield Mounting Medium with 4′,6-diamidino-2-phenylindole (ab10413; Abcam). PBS served as the negative control. A confocal laser scanning microscope (Leica TCS-SP8; Leica Microsystems, Wetzlar, Germany) was used to capture fluorescence images.

### 4.9. Statistical Analysis

Each experiment was performed at least three times. Results are expressed as the mean and standard deviation. One-way analysis of variance followed by the post-hoc Tukey–Kramer test or Bonferroni’s test was employed for multiple comparisons. Student’s *t*-test was used to compare the results of two groups. *p* < 0.05 was considered statistically significant.

## 5. Conclusions

Upregulation of miR-27a-5p and proinflammatory cytokines IL-6, IL-8, and MCP1 was induced in LPS-stimulated hDPCs, which was mediated through an NF-κB-dependent mechanism. Moreover, miR-27a-5p suppressed the expression of proinflammatory cytokines via NF-κB signaling by directly targeting TAB1, IRAK4, RELA, and FSTL1 in hDPCs stimulated with LPS. MiR-27a-5p-binding sequences in the TAB1 3ʹ-UTR were determined, and knockdown of TAB1 induced downregulation of proinflammatory mediator expression and NF-κB activity. Comparable observations were replicated in LPS-stimulated rat pulp tissues ex vivo. These results contribute to the understanding of the putative negative regulatory effects of miR-27a-5p, particularly those targeting the TAB1-NF-κB signaling pathway, on pulp inflammation.

## Figures and Tables

**Figure 1 ijms-25-09694-f001:**
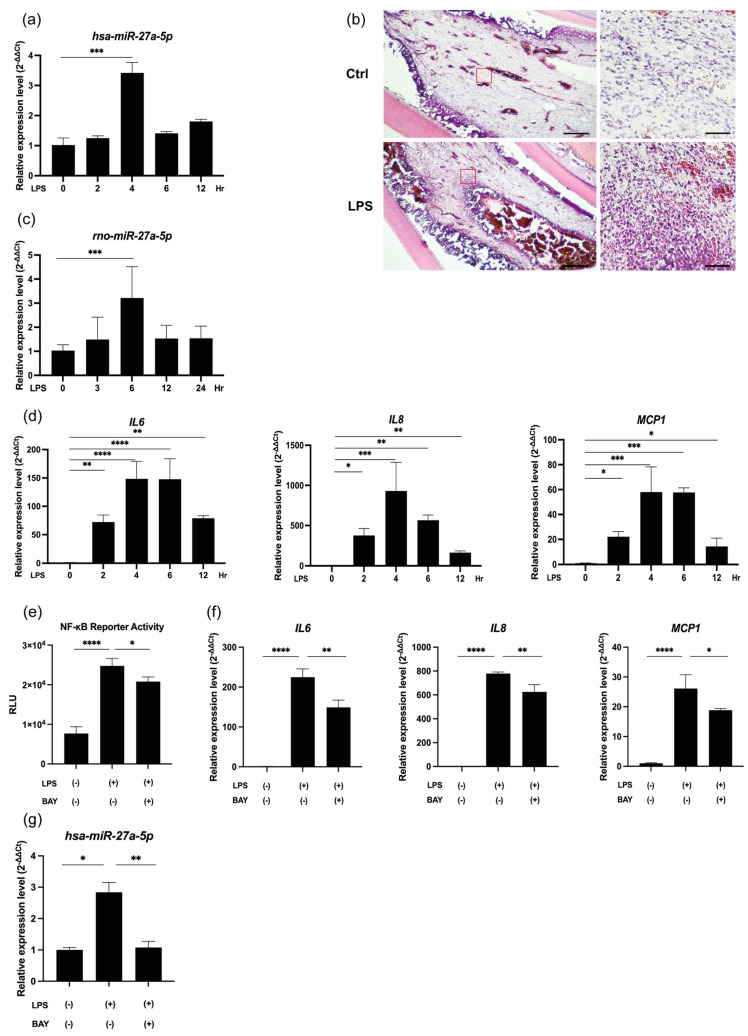
NF-κB signaling induces synthesis of miR-27a-5p and proinflammatory cytokines in LPS-stimulated hDPCs. (**a**) MiR-27a-5p expression was measured by RT-qPCR in hDPCs stimulated with 100 ng/mL LPS (mean ± SD, n = 4). (**b**) H&E staining indicated a typical inflammatory reaction characterized by blood vessel dilation and infiltration of inflammatory cells in LPS-induced inflamed rat pulp tissues (Scale bars: 400 μm in low magnification; 80 μm in high magnification). (**c**) MiR-27a-5p expression was measured by RT-qPCR (mean ± SD, n = 6) in LPS-induced inflamed rat pulp tissues. (**d**) mRNA expression of proinflammatory cytokines *IL-6*, *IL-8*, and *MCP1* under LPS stimulation (100 ng/mL) at various times was determined by RT-qPCR (mean ± SD, n = 4). (**e**) BAY 11-7085 (1 μM), an NF-κB inhibitor, was used to pretreat hDPCs for 2 h. LPS (100 ng/mL) was then used to stimulate the cells for 4 h. NF-κB reporter activity was measured by luciferase reporter assays (mean ± SD, n = 4). (**f**) mRNA expression of *IL-6, IL-8*, and *MCP1* with or without BAY 11-7085 pretreatment was measured by RT-qPCR. (mean ± SD, n = 4). (**g**) MiR-27a-5p expression levels with or without BAY 11-7085 pretreatment were evaluated by RT-qPCR (mean ± SD, n = 4). **** *p* < 0.0001; *** *p* < 0.001; ** *p* < 0.01; * *p* < 0.05. LPS: lipopolysaccharide; hDPCs: human dental pulp cells.

**Figure 2 ijms-25-09694-f002:**
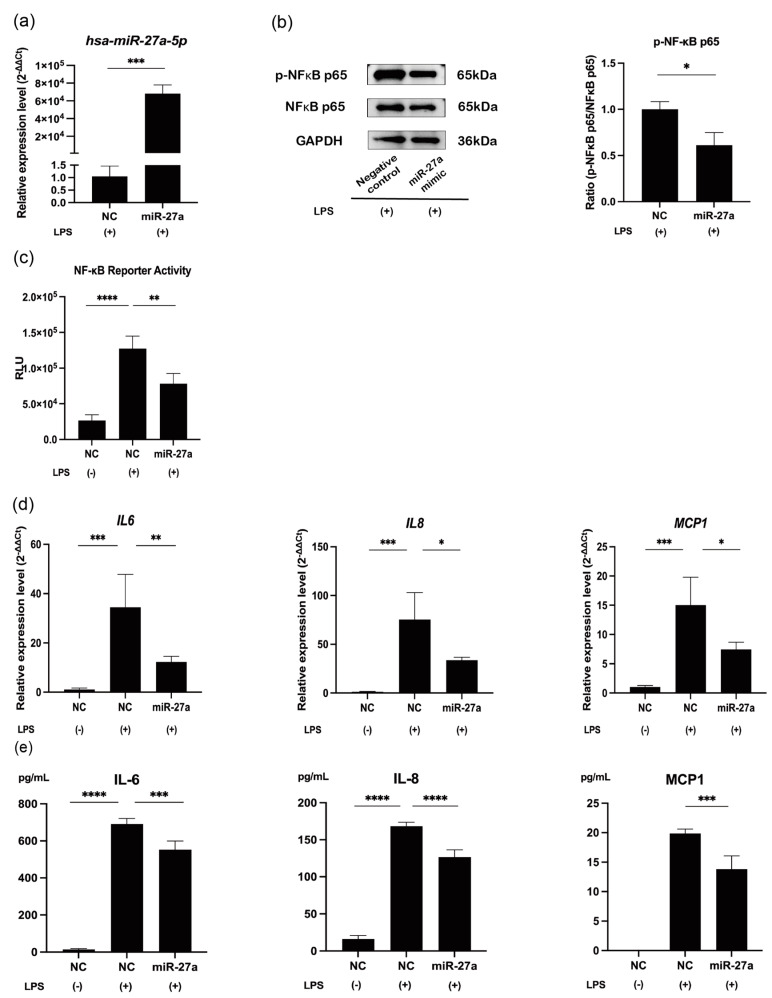
MiR-27a-5p mimic downregulates the expression of proinflammatory cytokines via the NF-κB pathway. (**a**) hDPCs were transfected with NC or the miR-27a-5p mimic for 24 h and then stimulated with LPS for 4 h. MiR-27a-5p expression was measured by RT-qPCR to verify the transfection efficiency (mean ± SD, n = 4). (**b**) Western blot analysis of NF-κB p65 (RELA) phosphorylation in LPS-stimulated hDPCs with GAPDH serving as the internal control (mean ± SD, n = 3). (**c**) NF-κB reporter activity was evaluated by luciferase reporter assays in LPS-stimulated hDPCs (mean ± SD, n = 4). (**d**) mRNA expression of *IL-6, IL-8*, and *MCP1* measured by RT-qPCR after miR-27a-5p overexpression in LPS-stimulated hDPCs (mean ± SD, n = 4). (**e**) Cytometric bead array of proinflammatory cytokines after miR-27a-5p overexpression in LPS-stimulated hDPCs (mean ± SD, n = 4). **** *p* < 0.0001; *** *p* < 0.001; ** *p* < 0.01; * *p* < 0.05. LPS: lipopolysaccharide; hDPCs: human dental pulp cells; NC: miRNA mimic Negative Control #1; miR-27a: miRNA mimic for miR-27a-5p.

**Figure 3 ijms-25-09694-f003:**
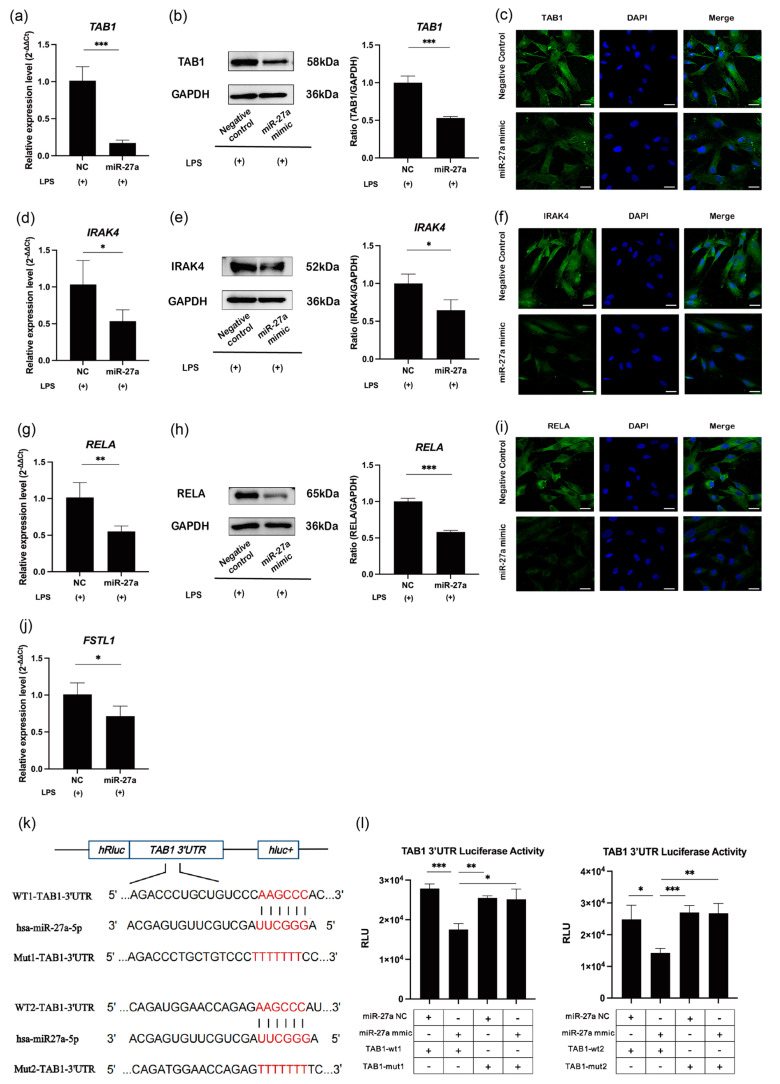
MiR-27a-5p overexpression downregulates NF-κB signaling via TAB1, IRAK4, RELA, and FSTL1 in LPS-stimulated hDPCs. (**a**,**d**,**g**,**j**) mRNA expression of TAB1, IRAK4, RELA, and FSTL1 measured by RT-qPCR after miR-27a-5p overexpression in LPS-stimulated hDPCs (mean ± SD, n = 4). (**b**,**e**,**h**) Western blot analysis of TAB1, IRAK4, and RELA after miR-27a-5p overexpression in LPS-stimulated hDPCs with GAPDH serving as the internal control (mean ± SD, n = 3). (**c**,**f**,**i**) hDPCs were stained for TAB1, IRAK4, and RELA by immunofluorescence after transfection with NC or the miR-27a-5p mimic under LPS simulation (Scale bars = 25 μm). (**k**) Schematic displaying a pairing relationship between miR-27a-5p and the TAB1 mRNA 3′-UTR. (**l**) Luciferase reporter activity of TAB1-WT 1/2 and TAB1-MUT 1/2 vectors in NC and miR-27a-5p groups was detected by a luciferase reporter assay (mean ± SD, n = 4). *** *p* < 0.001; ** *p* < 0.01; and * *p* < 0.05. LPS: lipopolysaccharide; hDPCs: human dental pulp cells; NC: miRNA mimic Negative Control #1; miR-27a: miRNA mimic for miR-27a-5p.

**Figure 4 ijms-25-09694-f004:**
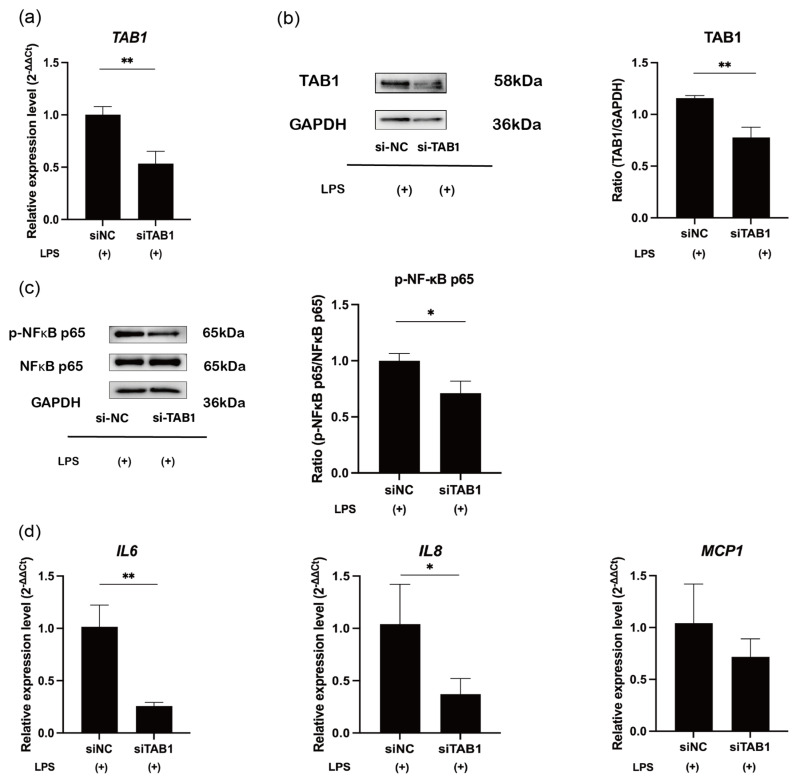
siTAB1 downregulates the expression of TAB1 and proinflammatory cytokines. (**a**) hDPCs were transfected with si-NC or si-TAB1 for 24 h and then stimulated with LPS for 4 h. mRNA expression of TAB1 was measured by RT-qPCR to verify the transfection efficiency (mean ± SD, n = 4). (**b**) Western blot analysis of TAB1 with GAPDH serving as the internal reference (mean ± SD, n = 3). (**c**) Western blot analysis of NF-κB p65 (RELA) phosphorylation in LPS-stimulated hDPCs with GAPDH serving as the internal control (mean ± SD, n = 3). (**d**) mRNA expression of *IL-6*, *IL-8*, and *MCP1* measured by RT-qPCR in hDPCs after si-ΤΑΒ1 transfection under LPS stimulation (mean ± SD, n = 4). ** *p* < 0.01; * *p* < 0.05. LPS: lipopolysaccharide; hDPCs: human dental pulp cells.

**Figure 5 ijms-25-09694-f005:**
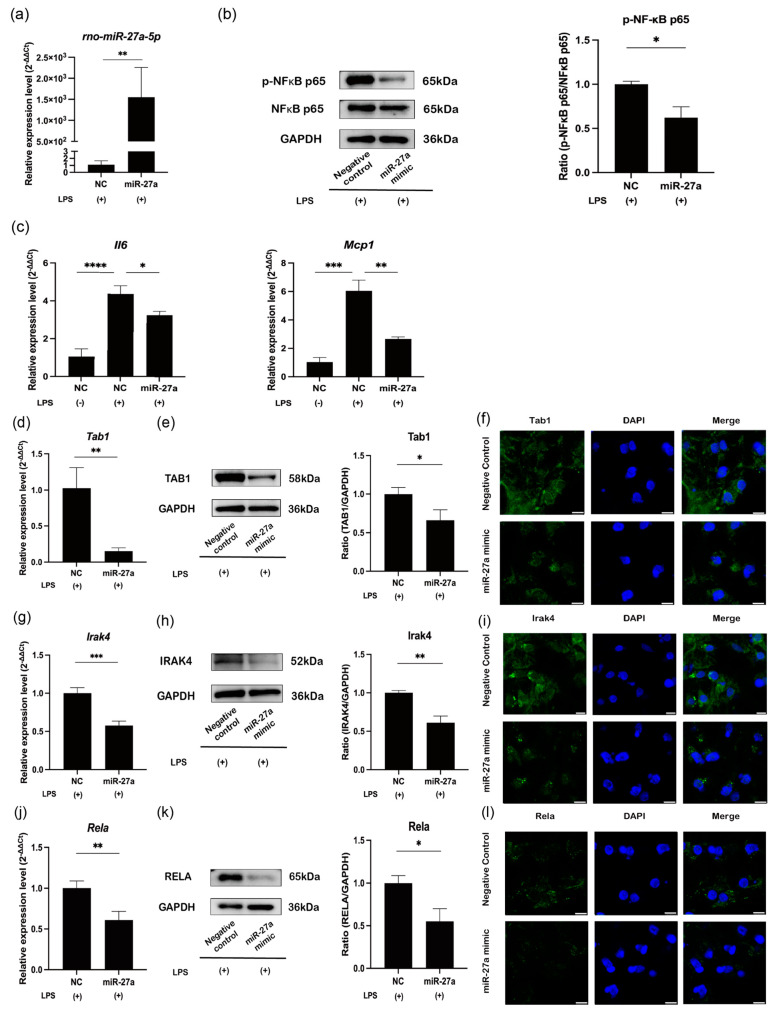
miR-27a-5p mimic downregulates proinflammatory cytokines via the NF-κB pathway in LPS-stimulated rat dental pulp tissue ex vivo. (**a**) Freshly extracted rat incisor dental pulp tissue was transfected with miR-27a-5p mimic or NC for 24 h and then stimulated with 200 ng/mL LPS for 4 h. MiR-27a-5p expression was measured by RT-qPCR to verify the transfection efficiency (mean ± SD, n = 4). (**b**) Western blot analysis of NF-κB p65 (RELA) phosphorylation with GAPDH serving as the internal control (mean ± SD, n = 3). (**c**) mRNA expression of *Il6* and *Mcp1* measured by RT-qPCR (mean ± SD, n = 4). (**d**,**g**,**j**) mRNA expression of Tab1, Irak4, and Rela measured by RT-qPCR (mean ± SD, n = 4). (**e**,**h**,**k**) Western blot analysis of Tab1, Irak4, and Rela with GAPDH serving as the internal control (mean ± SD, n = 3). (**f**,**i**,**l**) Immunofluorescence analysis of TAB1, IRAK4, and RELA. Scale bars = 5 μm. **** *p* < 0.0001; *** *p* < 0.001; ** *p* < 0.01; * *p* < 0.05. LPS: lipopolysaccharide; hDPCs: human dental pulp cells; NC: miRNA mimic Negative Control #1; miR-27a: miRNA mimic for miR-27a-5p.

**Figure 6 ijms-25-09694-f006:**
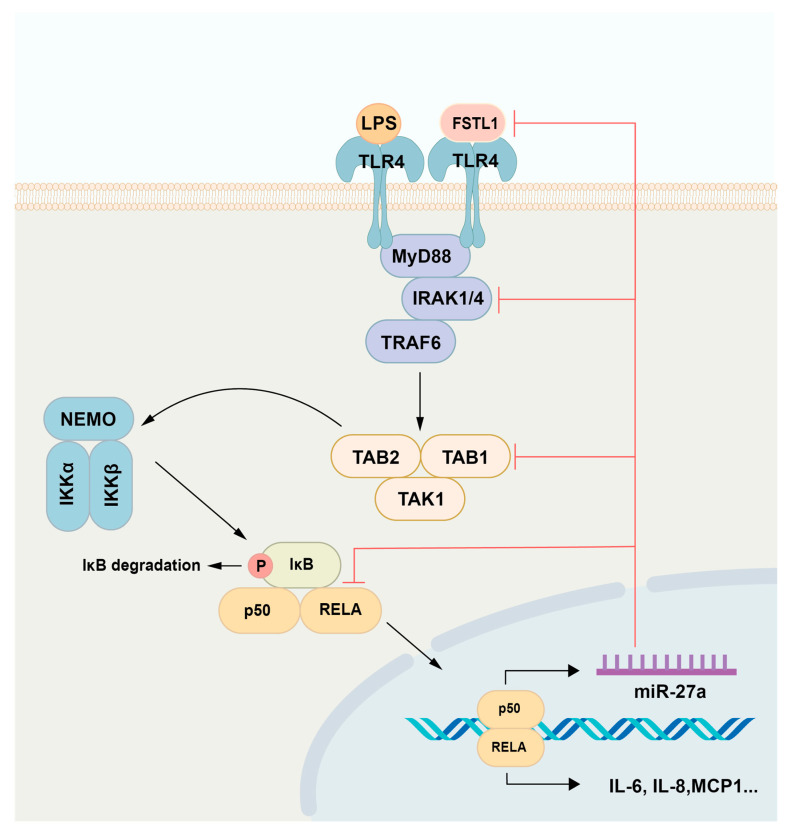
Schematic diagram of miR-27a-5p functions in hDPCs. LPS binding to TLR4 stimulates the NF-κB signaling pathway, which induces the synthesis of proinflammatory mediators (IL-6, IL-8, and MCP1) and miR-27a-5p in hDPCs. In turn, miR-27a-5p downregulates the expression of proinflammatory mediators by suppressing TAB1 and IRAK4, important intermediates of the NF-κB signaling pathway, RELA, a subunit of NF-κB, and FSTL1. LPS: lipopolysaccharide; hDPCs: human dental pulp cells.

**Table 1 ijms-25-09694-t001:** Sequences for primers used in RT-qPCR.

Gene	Forward	Reverse	Accession No.	Size, bp
<human>				
*ACTB*	5′-GTAGCACAGCTTCTCCTTAATGTCA-3′	5′-CTGACTGACTACCTCATGAAGATCC-3′	NM_001101.3	102
*IL6*	5′-TATACCTCAAACTCCAAAAGACCAG-3′	5′-ACAAGAGTAACATGTGTGAAAGCAG-3′	NM_000600.4	157
*IL8*	5′-TCAGTGCATAAAGACATACTCCAAA-3′	5′-TCTTCCATCAGAAAGCTTTACAATAA-3′	NM_000584.4	121
*MCP1*	5′-CACCTGCTGTTATAACTTCACCAAT-3′	5′-GTTGAAAGATGATAAGCCCACTCTA-3′	NM_002982.4	130
*TAB1*	5′-ATCCCTCAGTGCCAACTAAACC-3′	5′-GAAGATCCCAGTGCACAAGTCA-3′	NM_153497.3	137
*IRAK4*	5′-CGGAAATCTCTTTATCATCCGTGAG-3′	5′-GCACATATGTTGATGGTGTTATGGG-3′	NM_001351341.2	126
*RELA*	5′-TTCCAAGTTCCTATAGAAGAGCAGC-3′	5′-TCAAAGATGGGATGAGAAAGGACAG-3′	NM_021975.4	134
*FSTL1*	5′-CCATGACCTGTGACGGAAAGAAT-3′	5′-TTAGATCTCTTTGGTGCTCACTCT-3′	NM_007085.5	137
<rat>				
*Actb*	5′-GTAAAGACCTCTATGCCAACACAGT-3′	5′-GGAGCAATGATCTTGATCTTCATGG -3′	NM_031144.3	127
*Il6*	5′-TAAGGACCAAGACCATCCAACTCAT-3′	5′-AGTGAGGAATGTCCACAAACTGATA-3′	NM_012589.2	125
*Mcp1*	5′-CTAAGGACTTCAGCACCTTTGAATG-3′	5′-GTTCTCTGTCATACTGGTCACTTCT-3′	NM_031530.1	120
*Tab1*	5′-TAGTGTCTGCTTCTGTTAGATCCTG-3′	5′-AATCAGCTTCCTCATCAGAGTGAAA-3′	NM_001109976.2	134
*Irak4*	5′-CTGAACGTGCTTTGTCTTTAACAAG-3′	5′-GTTGAAATGAGCTCCGTACTAAGTG-3′	NM_001106791.2	121
*Rela*	5′-CTTTCTCAAGTGCCTTAATAGCAGG-3′	5′-TTCAGAGCTAGAAAGAGCAAGAGTC-3′	NM_199267.2	121

## Data Availability

The datasets that support the findings of the present study are available from the corresponding author upon reasonable request.

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
