# Peer review of "MicroRNA-27a-5p Downregulates Expression of Proinflammatory Cytokines in Lipopolysaccharide-Stimulated Human Dental Pulp Cells via the NF-κB Signaling Pathway"

_ijms, 2024, doi:10.3390/ijms25179694_

Round 1

Reviewer 1 Report

Comments and Suggestions for Authors

This manuscript is focused on the microRNA-27a-5p and its action to reduce LPS-induced inflammatory response in human dental pulp cells. The authors provide evidence to show that microRNA-27a-5p reduces pro-inflammatory response to LPS by a NF-kB mediated mechanism. The manuscript is well written, the work appears to be well carried out. there are some minor comments/questions.

1. The expression of miR-27a-5p was increased after LPS but the expression of cytokines was also increased. Its only with the transfection of of miR-27a-5p mimic that causes the reduction in pro-inflammatory cytokines. Please comment on this discrepancy.  

2. Size of Figure 1b needs to be increased as it is very difficult to see changes clearly..

3. The BAY compound reduced NF-kB activity by a small amount but completely abolished miRNA-27a-5p expression completely. The authors should comment on this unusual finding. 

Comments on the Quality of English Language

This is a well written manuscript.

Author Response

Comments 1: The expression of miR-27a-5p was increased after LPS but the expression of cytokines was also increased. Its only with the transfection of of miR-27a-5p mimic that causes the reduction in pro-inflammatory cytokines. Please comment on this discrepancy.  

Response 1: Thank you for your observation. LPS stimulation induces the activation of NF-kB signaling pathway, leading to the production of inflammatory mediators downstream, while simultaneously inducing the production of miR-27a. miR-27a does not directly block the NF-kB signaling pathway but rather suppresses the production of NF-kB signaling components. Additionally, the production of miR-27a increases over time. For these reasons, there is a time lag before miR-27a can effectively inhibit the NF-kB signaling pathway following the LPS stimulation and suppress the production of inflammatory mediators.

When miR-27a-5p mimic is introduced, it significantly elevates miR-27a-5p levels beyond the endogenous increase, leading to a more pronounced reduction in cytokines. This suggests that while endogenous miR-27a-5p may contribute to reducing inflammation, higher concentrations of miR-27a-5p achieved through mimic transfection more effectively suppress inflammatory cytokines, potentially through enhanced modulation of the NF-kB signaling pathway.

In summary, both endogenous and exogenous increases in miR-27a-5p have the potential to influence cytokine levels, but the effectiveness of this modulation appears to be more pronounced with mimic transfection due to higher miR-27a-5p levels.

Comments 2: Size of Figure 1b needs to be increased as it is very difficult to see changes clearly.

Response 2: Thank you for your feedback. I have enlarged Figure 1b to enhance visibility.

Comments 3: The BAY compound reduced NF-kB activity by a small amount but completely abolished miRNA-27a-5p expression completely. The authors should comment on this unusual finding. 

Response 3: We appreciate your insight into the observed effects of the BAY compound. The detailed mechanism by which BAY administration strongly suppresses miR-27a expression is not clearly understood. It is possible that factor(s) other than NF-kB are involved in the synthesis of miR-27a, and BAY may have inhibited these factor(s).

Reviewer 2 Report

Comments and Suggestions for Authors

The authors of this study sought to elucidate the role of microRNA-27a-5p in pulp inflammation, focusing on the expression  of inflammatory cytokines after LPS stimulation in human  pulp cells and ex vivo tissue samples from mice.

 The experimental design of this study was well developed, and the results of several molecular assays were able to support the authors' hypothesis.

However, I would like to clarify the following points (And I suggest the authors can include them in the discussion):

How can the results of this biological study be applied to the field of endodontics? I cannot find the link in the discussion section

Why did the authors choose this very specific molecule? Where did this story come from? Another paper with similar results and strategies was published in 2023 https: //pubmed.ncbi.nlm.nih.gov/37108595/.

 Why did the authors choose an ex vivo model?

 What are the advantages of this experimental design?

 Why did the authors not use an in vivo model having animals available?

Discussion Line 280, said (Figure 1a, c). Please remove these quotes from the figures during the discussion. They are unnecessary as they are explained in the results section and figure captions

The discussion seems to be very short and the authors mainly write about the results of this study. Although there is little information about microRN27a-5p, the authors could improve it by comparing these results to previous studies using similar research agents outside the field of dentistry. Additionally, it should be written in a way that the reader can understand the implications of this purely biological study.

Figure 6 should be better explained and discussed in the text.

Materials and Methods: Information on the reagent usage procedure is incomplete.

 For example: line 384: (FUJIFILM, Pure Wako Chemical), lack of country of origin and  many other similar cases in the section.

Comments on the Quality of English Language

Quality of English language is good. Minor changes might be done, but overall the readers can follow the content without major problems.

Author Response

Comments 1: How can the results of this biological study be applied to the field of endodontics? I cannot find the link in the discussion section

Response 1: Thank you for your valuable suggestion. We have now included a discussion on the application of our findings to the field of endodontics (Page 13, Line 329-337). We have addressed how the modulation of exosomal miRNA could impact other fields and endodontics therapy. The discussion now addresses the potential influence of miR-27a-5p in managing inflammatory responses and improving therapeutic strategies within endodontics.

Comments 2: Why did the authors choose this very specific molecule? Where did this story come from?

Another paper with similar results and strategies was published in 2023 https: //pubmed.ncbi.nlm.nih.gov/37108595/.

Response 2: Thank you for your question. In pulpal inflammation, various miRNAs are thought to form clusters and regulate the inflammatory response. We screened miRNAs whose synthesis from dental pulp cells increases upon LPS stimulation using an array (J Cell Physiol. 2019 Nov;234(11):21331-21341), and miR-21 (J Cell Physiol. 2019), miR-146b (Mol Sci. 2023 Apr 18;24(8):7433) and the current miR-27a were selected from the miRNAs whose expression increased in LPS-stimulated dental pulp cells.

Comments 3: Why did the authors choose an ex vivo model?

 What are the advantages of this experimental design?

 Why did the authors not use an in vivo model having animals available?

Response 3: Thank you for your question. The ex vivo experimental design allows precise manipulation of variables and isolates the effects of miR-27a-5p without systemic influences. The design also minimizes biological variability compared to in vivo models, leading to clearer insights into specific cellular mechanisms. Moreover, ex vivo models can provide quicker results as they often involve shorter experimental timelines compared to in vivo studies. In addition, ex vivo models are generally less resource-intensive and more feasible than in vivo studies. Based on these advantages, we chose the ex vivo experimental design as a useful model for investigating the direct effects of miR-27a-5p on inflammatory responses in dental pulp tissue.

Comments 4: Discussion Line 280, said (Figure 1a, c). Please remove these quotes from the figures during the discussion. They are unnecessary as they are explained in the results section and figure captions

Response 4: Thank you for your feedback. We have removed the references to (Figure ##) from the discussion section as suggested.

Comments 5: The discussion seems to be very short and the authors mainly write about the results of this study. Although there is little information about microRN27a-5p, the authors could improve it by comparing these results to previous studies using similar research agents outside the field of dentistry. Additionally, it should be written in a way that the reader can understand the implications of this purely biological study.

Response 5: Thank you for your insightful feedback. We have expanded the discussion section to include previous studies involving miRNA in other fields. We highlight the broader context of miR-27a-5p's role as an anti-inflammatory mediator and its potential applications beyond endodontics (Page 13, Line 329-337).

Comments 6: Figure 6 should be better explained and discussed in the text.

Response 6: Thank you for your feedback. We have added the explanation of Figure 6 in the text (Page 9, Line 225-228).

Comments 7: Materials and Methods: Information on the reagent usage procedure is incomplete.

 For example: line 384: (FUJIFILM, Pure Wako Chemical), lack of country of origin and  many other similar cases in the section.

Response 7: Thank you for your feedback. We would prefer to indicate the city and country of a company only at the first mention to avoid repetitions.